# Accuracy of haplotype estimation and whole genome imputation affects complex trait analyses in complex biobanks

Vivek Appadurai [1,2✉], Jonas Bybjerg-Grauholm [2,3], Morten Dybdahl Krebs[1,2], Anders Rosengren[1,2], Alfonso Buil [1,2], Andrés Ingason[1,2], Ole Mors[2,4], Anders D. Børglum [2,5,6], David M. Hougaard [2,3], Merete Nordentoft[2,7,8], Preben B. Mortensen[2,9,10], Olivier Delaneau [11], Thomas Werge [1,2] & Andrew J. Schork[1,2,12✉]

Sample recruitment for research consortia, biobanks, and personal genomics companies span years, necessitating genotyping in batches, using different technologies. As marker content on genotyping arrays varies, integrating such datasets is non-trivial and its impact on haplotype estimation (phasing) and whole genome imputation, necessary steps for complex trait analysis, remains under-evaluated. Using the iPSYCH dataset, comprising 130,438 individuals, genotyped in two stages, on different arrays, we evaluated phasing and imputation performance across multiple phasing methods and data integration protocols. While phasing accuracy varied by choice of method and data integration protocol, imputation accuracy varied mostly between data integration protocols. We demonstrate an attenuation in imputation accuracy within samples of non-European origin, highlighting challenges to studying complex traits in diverse populations. Finally, imputation errors can bias association tests, reduce predictive utility of polygenic scores. Carefully optimized data integration strategies enhance accuracy and replicability of complex trait analyses in complex biobanks.

[1] Institute of Biological Psychiatry, Mental Health Center Sankt Hans, Roskilde 4000, Denmark. [2] The Lundbeck Foundation Initiative for Integrative Psychiatric Research, iPSYCH, Aarhus, Denmark. [3] Danish Center for Neonatal Screening, Statens Serum Institut, Copenhagen, Denmark. [4] Psychosis Research Unit, Aarhus University Hospital - Psychiatry, Aarhus, Denmark. [5] Department of Biomedicine and Center for Integrative Sequencing, iSEQ, Aarhus University, Aarhus, Denmark. [6] Center for Genomics and Personalized Medicine, CGPM, Aarhus University, Aarhus, Denmark. [7] Mental Health Services in the Capital Region of Denmark, Copenhagen, Denmark. [8] Department of Clinical Medicine, Faculty of Health Sciences, University of Copenhagen, Copenhagen, Denmark. [9] NCRR - National Center for Register-Based Research, Business and Social Sciences, Aarhus University, Aarhus, Denmark. [10] CIRRAU - Centre for Integrated Register-Based Research, Aarhus University, Aarhus, Denmark. [11] Department of Computational Biology, University of Lausanne, Lausanne, Switzerland. [12] The Translational Genomics Research Institute, Phoenix, AZ, USA. ✉email: vivek.appadurai@regionh.dk; andrew.joseph.schork@regionh.dk

A recent appreciation for the polygenic nature of complex traits, with several small-effect risk loci scattered throughout the genome has revealed that genome-wide association studies (GWAS)[1,2] often employ hundreds of thousands of participants to identify trait-associated loci. Due to their cost-effective nature, genotyping arrays, that ascertain between 200,000 to 2 million single nucleotide polymorphisms (SNPs) in the human genome, have become the preferred technology for generating genetic data at such sample sizes. A key component of these studies is reference-based whole-genome imputation (imputation), which expands the number of markers that can be studied[3], in a two-step process. First, a collection of genotyped SNPs are organized into haplotype scaffolds (phased), relying on co-inheritance patterns of SNPs (i.e., linkage disequilibrium, LD). Known, untyped variants are then probabilistically imputed by matching these sparse scaffolds to more dense haplotypes from whole genome sequenced reference individuals[4]. This process results in a much larger pool of variants, thereby increasing GWAS power[5]. Importantly, it helps build a common set of SNPs for meta-analysis across cohorts genotyped on different arrays[6], and ensures sufficient overlap of SNPs between reference and target datasets for polygenic scoring (PGS)[7]. Various computational methods and reference datasets have been designed for this purpose. Research cohorts beginning with different marker sets, in diverse batches are often combined, even within a single population study.

State-of-the-art phasing methods, such as BEAGLE5[8], SHAPEIT4[9], and EAGLE2[10] use hidden markov model approaches built on the Li and Stephens model[11]. This model assumes that an individual's genome can be constructed as a mosaic of segments from haplotypes observed in the reference data or the study population, while accounting for additional factors such as recombination and de novo mutation rates. Current phasing methods differ in their computational approximations and data structures used for selecting the most informative haplotypes. Each phasing method further accepts user-defined parameters to choose the number of informative haplotypes, with a trade-off between accuracy, run times, and memory usage. While phasing methods have been improved over the years to scale computationally with large datasets such as the UK biobank[12], benchmarking is often performed in subsets of the 1000 genomes project[13], UK biobank, genome in a bottle dataset[14], or the GERA cohort[15]. To the best of our knowledge, the robustness of these methods has not been tested on input datasets with varying SNP density, target sample sizes, and missingness that can arise when integrating data generated on different genotyping platforms. It is important to empirically characterize the accuracy of phasing and imputation in such scenarios so that researchers can make informed choices when designing bioinformatics workflows to construct next-generation biobanks.

The predominant approach used by research consortia for analyzing samples genotyped on multiple arrays has been to phase and impute them separately, prior to meta-analyzing the results for GWAS[16,17]. However, the accuracy of phasing has been demonstrated to increase with increased sample sizes of reference and target datasets[18]. Moreover, for samples generated from recent population-scale biobanks (e.g., UK biobank[12], iPSYCH[19]), the number of study individuals is often much greater than the largest available haplotype reference. Haplotype sharing among study individuals and geographical variation in haplotype frequency imply these study haplotypes are as informative, if not more than published references for phasing[20]. Hence, there is intuitive reasoning to pool together as many samples as possible for phasing. In the UK Biobank study, where 500,000 participants were genotyped in 33 batches using two genotyping arrays, it was possible to phase and impute the entire study population together, leveraging the unprecedented sample size because the arrays used, the UK

Biobank Axiom array and the UK BiLEVE array, were closely matched (95% marker overlap). However, challenges arise in scenarios where genotyping involves different arrays with low marker overlap and there is currently insufficient guiding research.

Earlier studies on integrating cohorts genotyped on different arrays were on a much smaller scale, used earlier generations of methods, and focused on less diverse cohorts. Sinnott et al.[21] compared imputed allele frequencies in two groups of healthy European ancestry controls, genotyped on different arrays with only ~30% overlap. They observed a substantial type-I error rate, even at genome-wide significance, due to associations with the genotyping array. Retaining only the set of SNPs imputed at the highest quality reduced, but did not eliminate, these errors. Uh et al.[22] combined two datasets imputed from arrays with 60,000 overlapping markers into a union data set with high levels of missingness. GWAS across all good-quality imputed markers showed an inflation in test statistics that was higher than when restricting to the markers genotyped on both arrays or when only including subjects genotyped on one array. The inflation was reduced when extreme quality control (QC) was applied ($r^2$ quality metric >0.98). Johnson et al.[23] compared two approaches for integrating cases and controls genotyped on different arrays. They observed that imputing from the union of SNPs across arrays led to 0.2% of SNPs showing associations to genotyping arrays, while imputing from the intersection led to lower imputation accuracy, albeit without the same bias. These previous studies highlight challenges associated with integrating genotype data, including the important notion of a potential accuracy/bias trade-off, but do not provide a consensus path forward.

Pimental et al.[24] studied the biases introduced by imputation in the context of estimating direct genomic values in livestock, analogous to PGS in human genetics. They observed a bias in imputed genotypes towards the more frequent (major) allele in the reference panel that caused estimated genomic values to be shrunk towards the sample mean. This bias was more evident in traits with high heritability and when genomic values were estimated using imputation from less dense haplotypes. More recently, Chen et al.[25] studied the impact of different combinations of phasing and imputation methods on PGS and demonstrated that while PGS differ at an individual level, when computed using imputed genotypes rather than gold standard whole genome sequencing, the variation at cohort level is low, resulting in a less than 5 percentile change in individual PGS rank within the cohort. The impact of imputation on PGS in the context of data integration across cohorts has otherwise remained underexplored and given the attention PGS have recently received[26–29], exploring these concepts in modern, population-scale, human complex trait genetics applications is critical.

This study uses the Lundbeck foundation initiative for integrative psychiatric research (iPSYCH) case-cohort dataset with an initial 81,330 subjects genotyped on the Infinium Psych Chip v1.0 (Illumina, San Diego, CA USA) and an additional 49,108 subjects genotyped on the Illumina Global Screening Array v2.0 (Illumina, San Diego, CA USA) to evaluate four realistic protocols (Fig. 1) for data integration. We compare the phasing accuracy using SHAPEIT4.1.2, EAGLE2.4.1, BEAGLE5, and a consensus approach in truth sets derived from 124 parent-offspring trios that were genotyped on both arrays. To compare the resulting imputation quality, we randomly masked 10,000 SNPs prior to phasing and included 10 whole genome sequenced samples from the Personal Genomes Project—UK cohort[30] (Supplementary Note 1) down-sampled to the SNPs in each cohort. Imputed genotypes were then compared to these truth sets to assess the loss of information in imputed data. It is known that current haplotype references are skewed towards individuals of European ancestry, hence we utilized the diverse ancestry composition of iPSYCH (Supplementary Table 1) to assess the quality of phasing

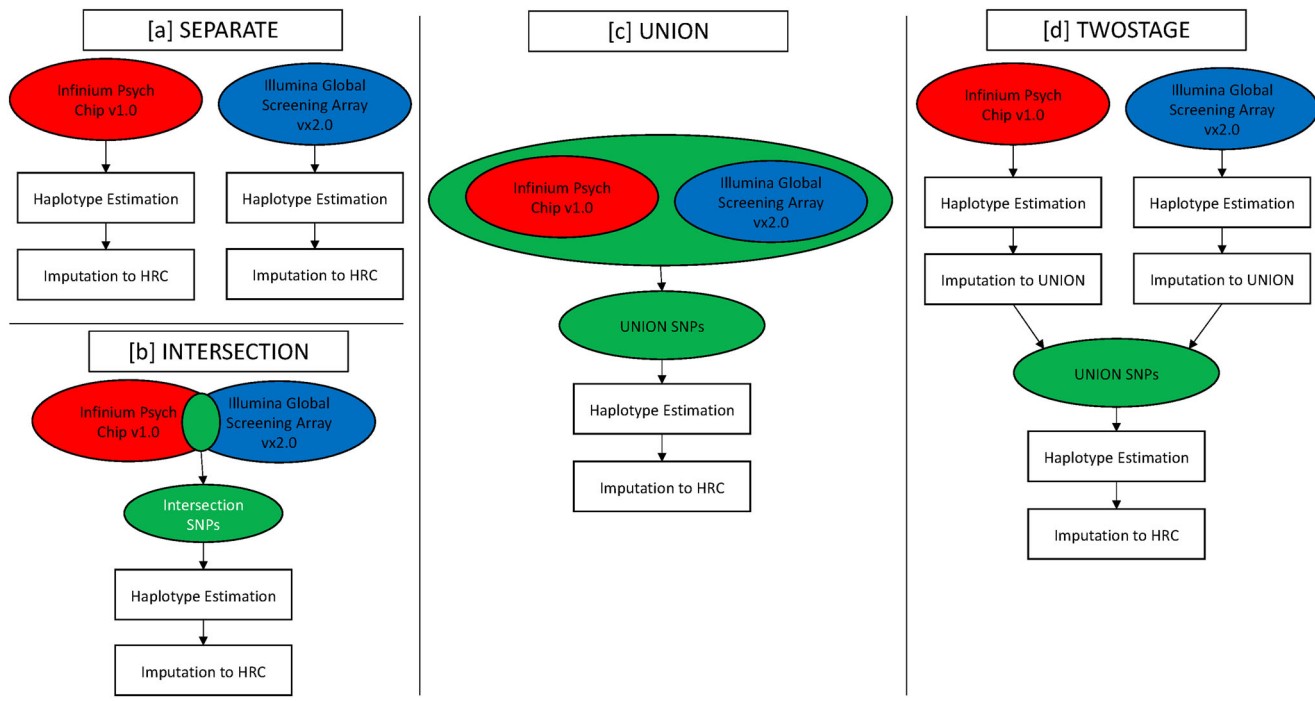

**Fig. 1 The four data integration protocols evaluated in the study. a** Shows the separate protocol where the cohorts genotyped on each array (Infinium psych chip v1.0 in red fill, Illumina global screening array 2.0 in blue fill) are phased and imputed separately. **b** Shows the intersection protocol where the two cohorts are merged to include only SNPs in common to both genotyping arrays (highlighted in green fill) prior to phasing and imputation. **c** Shows the union protocol where the two cohorts are merged to include SNPs genotyped on either array (highlighted in green fill) and the resulting dataset with missingness is phased and imputed. **d** Shows the two-stage protocol where the haplotypes obtained from the separate protocol are initially imputed to the markers in the union protocol (highlighted in green fill), prior to a second stage of phasing before the cohorts are split back to the original sets of genotyped SNPs after which imputation to the full reference panel is performed.

and imputation in non-European and admixed individuals. Finally, using a simulated quantitative trait, we explore the impact of phasing and imputations across data integration scenarios on GWAS and PGS.

## Results

**Phasing accuracy**. Phasing accuracy was measured using Switch Error Rates (SER, Methods) with three methods, two-parameter settings each, and a consensus set across four data integration protocols (Fig. 2a, Supplementary Data 1). Our results show that phasing accuracy depends on the data integration protocol, phasing methods and associated parameters, target sample size, genotyped SNP density in the target, rate, and structure of genotype missingness. In general, the two-stage protocol, which leverages the largest possible sample size and density of SNPs, with no missingness, shows consistently high accuracy across all phasing methods (SER = 0.17–0.55%). The intersection protocol, which also leverages the largest sample size, albeit with lowest SNP density, proves the least accurate (SER = 0.38–1.04%). The ranking of the protocols was generally consistent across methods, except for the union, which achieved the lowest overall SER with BEAGLE5 at parameter value, phase-states = 560. The union was also the worst-performing protocol when taking consensus haplotypes across all three methods (SER = 0.61% at default parameters), suggesting the genotype missingness introduced by this protocol causes systematic phasing errors that are reproduced across tools.

In protocols involving little to no genotype missingness (i.e., not Union), BEAGLE5 and SHAPEIT4.1.2 show similar accuracy, outperforming EAGLE2.4.1 across integration methods and parameters. The union was again a point of departure from the trends, with BEAGLE5 performing better (SER = 0.17%) on the

union and SHAPEIT4.1.2 performing better on the two-stage (SER = 0.17%). This implicates genotype missingness for phasing performance, suggesting that BEAGLE5 handles this more robustly than SHAPEIT4.1.2. When considering the two-stage protocol, which we hypothesized could mitigate initial missing genotypes, SHAPEIT4.1.2 performed similarly to BEAGLE5 on the union (and better than on the two-stage), suggesting, modulo initial missingness, SHAPEIT4.1.2 may have at least as good a phasing algorithm as BEAGLE5.

Comparing the phasing accuracy across chromosomes within each method and data integration protocol reveals that phasing accuracy follows the number of SNPs per centimorgan in the target dataset, with denser chromosomes showing lower SER (Supplementary Figure 1, Supplementary Data 2). We also observe that EAGLE2.4.1 and BEAGLE5 produce more accurate estimates in Cohort 2012 where the sample size is higher and SNP density is sparser whereas SHAPEIT4.1.2 produces more accurate estimates in Cohort 2015i where the SNP density is higher and the target sample size is comparatively smaller. As mentioned above, the worse performance of SHAPEIT4.1.2 and EAGLE2.4.1 on the union as opposed to the two-stage highlight the sensitivity to initial missing genotypes. These results show the necessity for benchmarking the robustness of phasing methods in less-than-ideal conditions, specific to study cohorts, prior to deploying them in such untested scenarios.

**Imputation accuracy**. The accuracy of imputations derived from each set of haplotype scaffolds (i.e., from each tool, parameters, and data integration protocol set) are presented in Fig. 2b and Supplementary Data 3. Variability in imputation accuracy stems more from the choice of data integration protocol, rather than the choice of phasing method or parameters. Since all methods

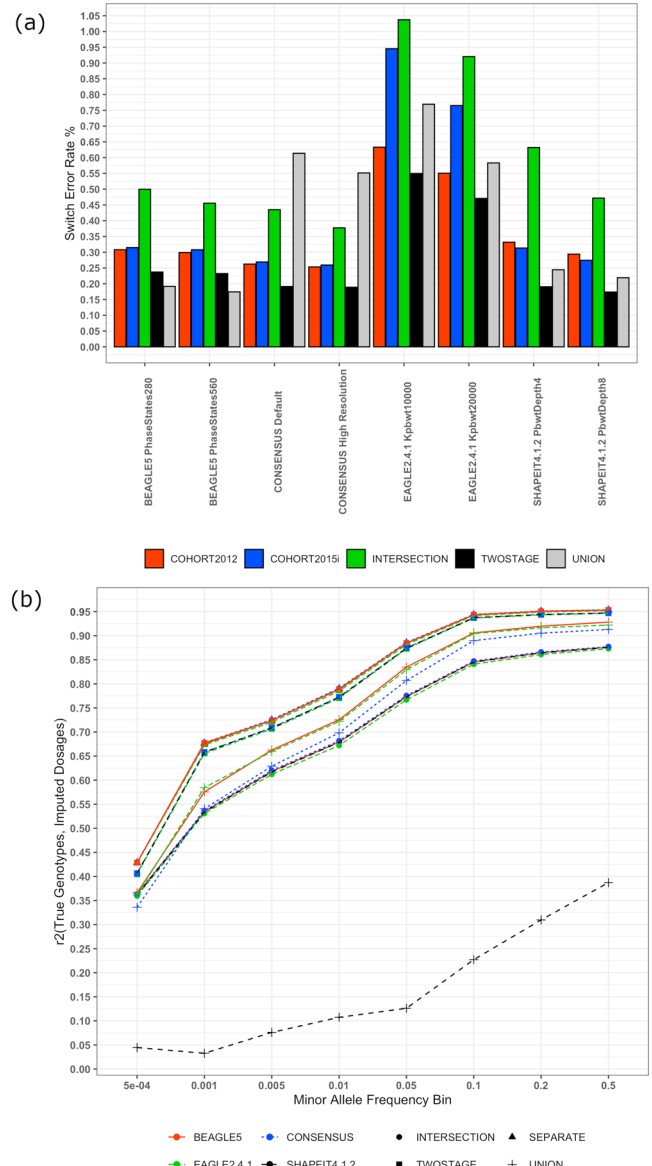

**Fig. 2 Phasing and imputation accuracy vary across state-of-the-art tools and data integration protocols. a** Shows the accuracy in switch error rate percentage of phasing across the three tools at two parameter sets each and a consensus approach taking the majority haplotype at each locus from the three tools at two parameter sets each across all four data integration protocols (cohort 2012 within the separate protocol in red bars, cohort 2015i within the separate protocol in blue bars, intersection protocol in green bars, two-stage protocol in black bars and union protocol in gray bars). Default parameters are SHAPEIT4.1.2 pbwt-depth = 4, BEAGLE5 phase-states = 280, EAGLE2.4.1 Kpbwt = 10000. High-Resolution parameters are SHAPEIT4.1.2 pbwt-depth = 8, BEAGLE5 phase-states = 560, EAGLE2.4.1 Kpbwt = 20,000. The switch error rates were computed within 124 trio offspring by comparing the computationally assigned phase to the mendelian transmission from known parental genotypes at the heterozygous loci common to both genotyping arrays. **b** Shows the imputation accuracy ($r^2$) within each data integration protocol (intersection protocol denoted by circles, separate protocol denoted by triangles, two-stage protocol denoted by squares, union protocol denoted by plus symbols), and choice of phasing tool (Beagle5 in red, Eagle2.4.1 in green, Shapeit4.1.2 in black, consensus across all three tools in blue) at different minor allele frequency bins across the 10,000 SNPs common to both genotyping arrays that were masked prior to phasing.

process data in variant call format (VCF), this renders the choice of phasing method less relevant if the end goal is to attain the most accurate missing data imputation. The highest imputation accuracy is obtained when the cohorts are phased separately, with the $r^2$ between true masked genotypes and imputed dosages varying between 0.43 at rare (MAF < 0.005) and 0.95 at more common (0.2 < MAF ≤ 0.5) SNPs. This trend is consistent across haplotypes generated by all methods. The added bioinformatics effort aimed at enhancing sample size without missingness with the two-stage protocol did not yield a higher imputation accuracy than the separate protocol. At the minor allele frequency bin, 0.01 < MAF ≤ 0.05, using haplotypes phased by BEAGLE5, both approaches show identical accuracy ($r^2 = 0.88$) (Supplementary Data 3). Imputation accuracy is degraded when using the inter-section protocol with an attenuation between 8.4–13.6% at common and 13.9–18.6% at rare SNPs as compared to the separate protocol, highlighting the drop in phasing accuracy at low target SNP density carrying over to imputation performance.

Haplotypes estimated by SHAPEIT4.1.2 in the union protocol are an outlier and resulting imputations are of noticeably poorer quality compared to haplotypes obtained from other methods. Phasing in the presence of missingness is itself a two-step process, where each phasing method makes a rough imputation of missing data prior to constructing haplotypes. If this data is not overwritten during imputation, the pre-phasing imputation algorithm implemented by SHAPEIT4.1.2 could be the reason for problems with the union protocol. This becomes more credible when considering the imputation accuracy obtained from the two-stage protocol using SHAPEIT4.1.2, where the attenuation is mitigated. The pattern of results described above is replicated in the PGP-UK samples (Supplementary Figure 2a, b).

A comparison of imputation accuracies between Cohort2012 and Cohort2015i within the separate protocol using the PGP-UK samples (Supplementary Figure 2c, Supplementary Data 4) shows higher imputation accuracy in Cohort2015i, imputed from a higher SNP density as compared to Cohort2012 with a larger sample size with a difference as high as 6.7% at MAF ∈ (0.1, 0.05]. This finding is important because it emphasizes a trade-off between sample size and SNP density and, with modern sample sizes, perhaps SNP density should be emphasized. Enhanced parameters that showed higher phasing accuracy do not seem to substantially increase imputation accuracy (Supplementary Figures 2a, b). Taken together, these results show that imputation performance suffers when merging cohorts genotyped on different arrays prior to phasing, and choice of phasing method is less relevant than the data integration protocol.

Previous studies have suggested lower variability in imputation accuracy as a function of imputation (versus phasing) tools[31], a result we confirmed in limited comparison of BEAGLE5.1[8], Minimac4[32], and Impute5[31] is presented in Supplementary Note 2, Supplementary Figure 3 and Supplementary Data 5.

**Imputation accuracy in non-European and admixed samples.** It is known that GWAS results and subsequent PGS constructed from them do not generalize well across populations[33]. This is typically attributed to inaccuracies in the estimation of SNP effect sizes (i.e., per SNP beta) due, e.g., to variable LD across populations[34]. However, if non-European haplotypes are underrepresented in either reference or target datasets, imputed genotypes in these individuals may be of lower quality, and errors in the genotypes themselves could be contributing to the generalization problems of GWAS. Imputation accuracy was estimated in non-European and admixed iPSYCH samples, grouped according to the birthplace of the proband's parents (Fig. 3, Supplementary Data 6). Within the separate protocol, Individuals born to non-Scandinavian European

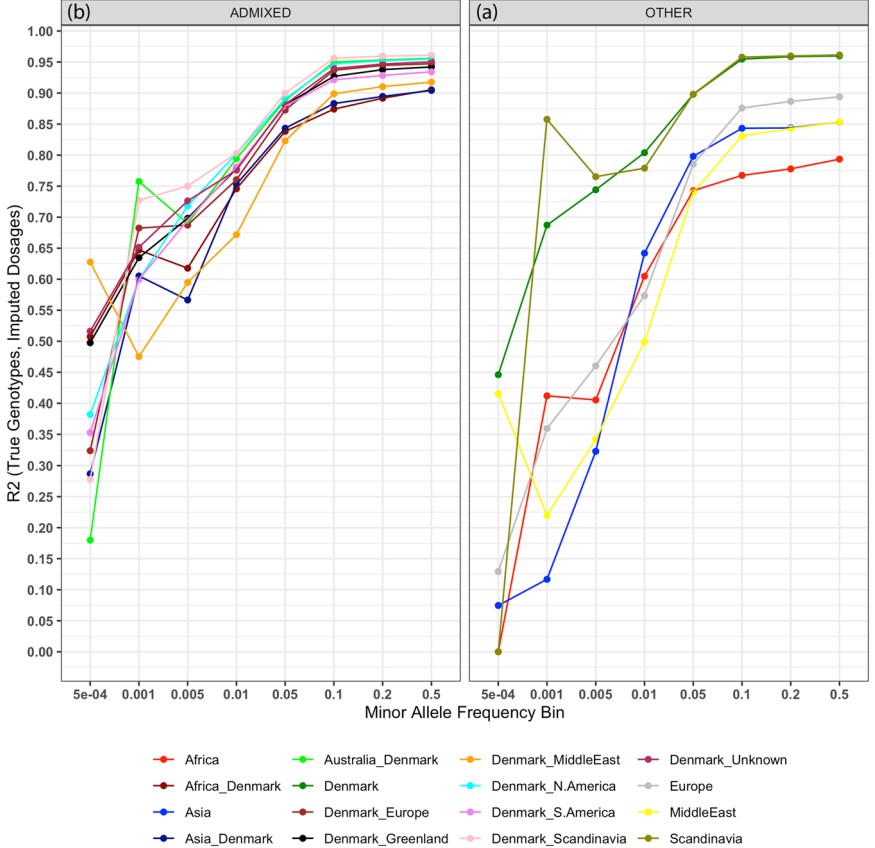

**Fig. 3 The accuracy of imputation varies extensively by parental birthplace. a** Shows the imputation accuracy ($r^2$) in iPSYCH samples grouped by parental birthplace as ascertained from the Danish civil registers at different minor allele frequency bins within the 10,000 SNPs common to both genotyping arrays, masked prior to phasing. (Both parents were born in Africa: Red, Asia: Blue, Denmark: Dark Green, Scandinavia non-Danish: Light Green, Europe non-Danish or Scandinavian: Gray, Middle East: Yellow). **b** Shows the imputation accuracy ($r^2$) in admixed samples where at least one parent was born in Denmark (Non-Danish parent born in Africa: Maroon, Asia: Blue, Australia: Fluorescent Green, Europe non-Danish or Scandinavian: Brown, Greenland: Black, Middle East: Orange, North America: Cyan, South America: Pink, Scandinavia non-Danish: Lavender, Unknown: Purple). All imputations were performed using a separate protocol, haplotype estimation was performed using BEAGLE5 phase-states = 560, imputations were performed using BEAGLE5.1 with the HRCv1.1 as the reference.

parents had lower imputation accuracy (7.07–12.58%) than those with both parents born in Denmark. These effects were larger for individuals with both parents born in Asia (11.1–11.2%), Africa (17.37–17.48%), or Middle East (11.2–17.7%). The attenuation in imputation accuracy within admixed individuals is comparatively lower, varying between 4.47 and 8.56% as compared to individuals with both parents born in Denmark. These results, as expected, suggest that imputation accuracy varies by ancestry and introduces a systematic loss of information in the genotypes of non-Europeans and this effect is more pronounced when employing less optimal data integration protocols (Supplementary Figure 4, Supplementary Data 6).

**Impact on PGS**. Imputations can contribute to sub-optimal predictive performance of PGS by introducing measurement error in the genotypes used in the discovery GWAS as well as in the target genotypes used for scoring. To empirically compare attenuation in predictive performance due to imputation errors at each stage, we utilized the independent sample ascertainment of the two iPSYCH cohorts, employing iPSYCH2012 as the discovery cohort while predicting in iPSYCH2015i and vice-versa. To estimate the impact of imputation errors in discovery GWAS, we performed a GWAS using PLINK with a simulated continuous phenotype as the outcome, regressed against the imputed dosages at the 10,000 masked SNPs in one iPSYCH cohort, and calculated

PGS by summing the product of the obtained SNP effect sizes with true genotypes at the 10,000 masked loci in the other iPSYCH cohort. To estimate the impact of imputation errors in target genotypes, we performed GWAS using true genotypes in one iPSYCH cohort, while calculating PGS in the other cohort as the running sum of the product of per SNP effect sizes with imputed dosages at the 10,000 masked SNPs (Methods). The genomic control coefficients, a measure of the power of the discovery GWAS arising from each data integration protocol are presented in Supplementary Note 3, Supplementary Table 2 and Supplementary Figure 5.

The results of the PGS analysis (Fig. 4a; Supplementary Data 7), suggest that the impact of imputation errors is more severe when present in target genotypes used for scoring than when they are present in the genotypes used for discovery GWAS. When using imputed dosages from iPSYCH2012 in a discovery GWAS and scoring using true genotypes in iPSYCH2015i, the only noticeable attenuation in the variance explained by PGS occurs in the intersection protocol, diminishing to 0.26 as compared to 0.27 in the ideal scenario, where both GWAS and scoring use true genotypes. On the other hand, when the discovery GWAS used true genotypes and scoring was done using imputed dosages, there is an attenuation in variance explained by PGS across all four imputation protocols, increasing in severity from the separate protocol ($r^2 = 0.26$) to the intersection protocol

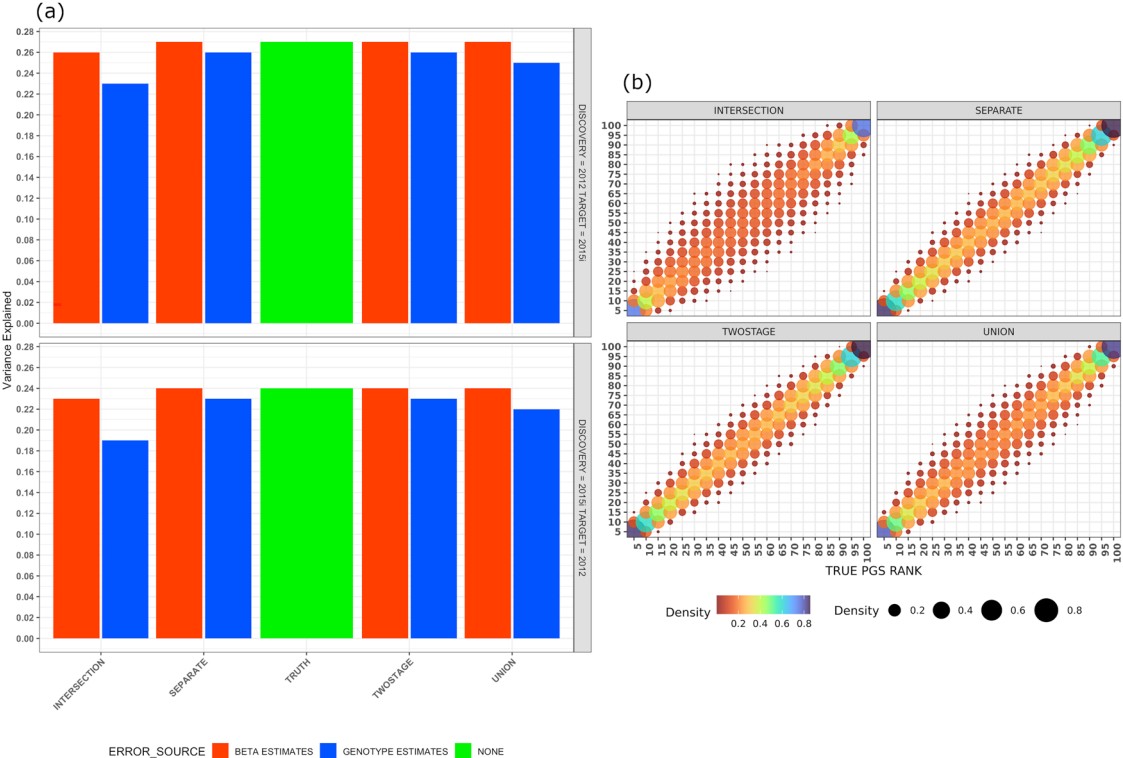

**Fig. 4 The performance of polygenic scores is attenuated when using imputed data, which affects both population-level measures such as variance explained in a simulated phenotype, and in individual-level metrics, such as rank concordance when compared to true genotypes. a** Shows the attenuation in variance explained by a PGS when an error introduced by using imputed dosages resides in either the discovery stage or in scoring. The green bars are the reference, where true genotypes were used for both discovery and scoring. Red bars indicate that imputed dosages were used in discovery, while blue bars indicate that imputed dosages were used for scoring. **b** Shows the rank concordance between individuals in iPSYCH2015i ranked according to their PGS calculated using true genotypes on X axis and imputed dosages from each data integration protocol, across the four figure panels on the Y axis. The rank concordance with the truth set is higher (Red: Low, Blue: High in heatmap gradient) when employing the separate or two-stage protocols as compared to union or intersection protocols.

($r^2 = 0.23$). This demonstration of reduced predictive capability of PGS in presence of imputed genotypes is in line with animal breeding studies[24] and highlights an additional challenge when projecting PGS across populations.

Another application of PGS is to prioritize individuals in top quantiles of a PGS distribution for monitoring and intervention. To investigate the effect of imputation accuracy on such an application, individuals in iPSYCH2015i were ranked into percentiles for the simulated continuous trait based on a PGS calculated using SNP effects with iPSYCH2012 as reference GWAS, true genotypes, and imputed dosages across all four data integration protocols, in turn accounting for the genetic dose. The results (Fig. 4b, Supplementary Data 8) are consistent with prior work[25] showing a discrepancy in individual rank that is higher in the middle percentiles and lower in the more actionable top percentiles of the PGS distribution. The discordance in individuals in the top percentiles between PGS constructed by true genotypes and imputed dosages is, however, much higher than the 5% previously reported. The overlap in the proportion of individuals ranked in the top 5 percentiles of PGS using true genotypes and imputed dosages is highest in both cohorts when employing either the separate (85%) or two-stage (86%) protocol and lowest when using the intersection (75%) protocol. Our results suggest that using imputed genotypes for the discovery GWAS stage of constructing a PGS might be more robust than when incorporating imputed genotypes into the scoring stage. The attenuation in predictive performance, and discordance of individuals in the actionable percentiles of PGS follow the quality of imputed dosages obtained from each data integration protocol, showcasing the importance of optimal data integration for genetic

prediction and may have special relevance for cross-ancestry PGS applications, where errors due to imputation are systematically larger.

**Batch effects**. Association studies were performed comparing genotypes and imputed dosages at the masked SNPs from all four data integration protocols in unrelated controls of iPSYCH2012 and 2015i of a homogenous genetic origin with the genotyping array as the outcome (see Methods). The resulting genomic inflation factor in test statistics across different thresholds for imputation quality is shown in Fig. 5a, Supplementary Data 9. The number of SNPs used in the association tests at each imputation quality threshold is shown in Fig. 5b, Supplementary Data 9.

The baseline for the inflation observed by comparing the genotyped SNPs in controls is $\lambda_{gc} = 1.05$. No inflation is observed when comparing SNPs imputed in both iPSYCH2012 and iPSYCH2015i using the intersection protocol, while test statistics are most inflated when using the union protocol. Using the separate and two-stage protocols, inflation is reduced at high thresholds of BEAGLE imputation $r^2$, but not eliminated. For example, in the separate protocol, with SNP imputation quality filter, DR2 > = 0.9, the $\lambda_{gc} = 1.13$ when comparing SNPs genotyped in iPSYCH2012 to SNPs imputed in iPSYCH2015i, and $\lambda_{gc} = 1.18$ when comparing SNPs imputed in iPSYCH2012 to SNPs genotyped in iPSYCH2015i and $\lambda_{gc} = 1.1$ when comparing SNPs imputed in both. At this threshold, 22% of the imputed SNPs are excluded. This analysis suggests that imputations performed from different genotyped backbones, which result in genotyped SNPs being compared to

imputed SNPs, will contain batch artifacts that can be difficult to remove by standard SNP exclusion, which might also be complicated, due to a lack of robustness of imputation quality metrics under different data integration protocols (Supplementary Note 4, Supplementary Figure 6).

## Discussion

As the cost of genotyping drops, the burden of complex trait analysis is moving away from genotyping requisite participants and towards storage, computational requirements, and the bioinformatics expertise to integrate and analyze such datasets[35]. Phasing and imputation have somewhat remained a black box in bioinformatics pipelines with researchers having the opportunity to avail themselves of services like the Michigan imputation server[32.] to reduce the computational burden of data preparation. However, privacy stipulations governing datasets generated through national biobanks might prohibit the use of such services. The benchmarking work presented in this study stresses the importance of making an educated choice of data integration protocols that could introduce a trade-off among peculiarities such as a sparse marker set, small sample size, high missingness in the input dataset, or the potential of batch artifacts.

The benchmarking of imputation accuracy presented in this study replicates previous findings[23], suggesting imputation from the intersection of markers when incorporating samples genotyped on multiple arrays leads to a loss of accuracy while imputation from the union of the markers leads to spurious associations with genotyping arrays[23]. Consistent with our hypothesis that the phasing accuracy could be improved by increasing the target sample size by jointly analyzing the two cohorts (by either the union or two-stage protocol) we did observe a drop in SER. However, these improvements did not result in improvements in imputation accuracy, likely reflecting that the phasing tools were not developed with this type of systematic missingness in mind. Until software that can leverage this apparent potential for improvement in phasing accuracy are available, our results suggest that phasing and imputing separately results in equivalent or better imputation accuracy. The higher phasing and imputation accuracies, PGS performance in the sub-cohort of iPSYCH individuals genotyped using the Illumina Global Screening Array, enriched with more common markers as compared to the sub-cohort imputed using the Infinium Psych Array, enriched for rare markers with prior associations to psychiatric phenotypes, suggests that when faced with a choice, it might be more beneficial to prioritize genotyping arrays with more common markers that overlap more with the content of haplotype reference panels. Analysis pipelines and methods focusing on common disease research, rely on established high-quality SNP sets, such as HapMap3, and use thresholds to exclude rare markers during QC, effectively rendering them useless for such applications.

Imputed data will contain non-random errors, especially in presence of systematic missingness, as can be the case when genotyping of samples is performed in batches and over time. Therefore, it is critical to consider the sensitivity of any analysis performed on these datasets. Technical artifacts in the genotype generation process are one of the sources of poor performance of PGS across cohorts[27]. While the attenuation introduced in PGS performance and the discordance of individual rank in different percentiles of the risk distribution when PGS are calculated using imputed data as compared to genotyped SNPs has received attention in animal breeding studies, this remains under-researched in human populations. As one of the clinically informative uses of PGS lies in selecting a subset of individuals in the actionable risk percentiles of a PGS distribution[26,27], errors introduced during phasing and imputation could have a sizable impact on genetic risk profiling—especially when data is acquired over time and according to different protocols. The

**Fig. 5 Inflation of test statistics shows type-I errors in association studies with imputed data across the four data integration protocols as compared to genotyped variants. a** Shows the inflation in test statistics represented using lambda genomic control when performing an association test at each of the 10,000 SNPs common to both genotyping arrays masked prior to phasing. Controls of a homogeneous genetic origin were compared between the iPSYCH2012 and iPSYCH2015i cohorts with the genotyping array as the outcome at different thresholds of post-imputation quality control across the four different data integration protocols (Array genotypes: black bars, intersection protocol: red bars, separate protocol: blue bars, union protocol: yellow bars, two-stage protocol: green bars). The dotted horizontal black line indicates the baseline $\lambda_{gc}$ when the association test was performed using true genotypes from both arrays. Haplotypes were phased using BEAGLE5 phase-states = 560, imputations were done using BEAGLE5.1 with the HRCv1.1 as the reference. **b** Shows the number of SNPs left after each threshold of post-imputation quality control across the four data integration protocols.

presence of spurious associations with genotyping arrays when comparing allele frequencies of genotypes and imputed dosages between cohorts as demonstrated in this study shows the need to pick stringent QC thresholds for GWAS. As stringent filtering might reduce the power due to exclusion of a majority of imputed SNPs, other approaches such as including the genotyping array as a covariate in regression models or as a fixed effect in linear mixed models need to be further investigated.

Haplotype reference panels employed for phasing and imputation are skewed towards Europeans and the evaluation of imputation accuracy within iPSYCH individuals, grouped by parental birthplace shows differentially worse accuracy in non-Europeans, stressing the need for reference panels with a more genetically diverse catalog of haplotypes, if genotyping arrays and imputation are to be used in precision medicine initiatives in a fair and equitable manner[33,34]. While considerable attention has been paid to the lack of PGS portability between populations due to less informative SNP effects, less attention has been paid to imputation quality in non-European populations, which introduces an additional source of error, not only in PGS but also in GWAS within these populations. While our comparisons held the reference population constant to the largest set of haplotypes that are currently publicly available, testing the imputation performance with varying references would also be informative. There has been demonstrable improvement in imputation accuracy for individuals of Hispanic/Latin and African descent using the NHLBI Trans-Omics for Precision Medicine whole genome sequenced reference panel[36], but it is currently only available through an imputation server, rendering its usage prohibitive for studies with data privacy stipulations.

It is important to acknowledge caveats associated with this study, that it studies the outcomes when the chosen genotyping arrays are very dissimilar, and the two cohorts being studied do not differ by an order of magnitude. Therefore, there can exist scenarios, where a data integration protocol such as union or two-stage might prove necessary to enhance the number of informative study haplotypes if one of the two cohorts is much smaller than the other. Similarly, if the overlap of SNPs is a magnitude higher, it might prove beneficial to prioritize the dramatic reduction in batch effects and simplicity of an intersection protocol. Such empirical comparisons can only be made by iterating across sample sizes and marker densities, across cohorts, in a more broadly orchestrated study.

In conclusion, this study demonstrates four different ways of integrating data genotyped on multiple arrays with sparse marker overlap. Care should be applied when integrating datasets and building biobanks for precision medicine initiatives, as improper treatment can hurt PGS performance, introduce batch artifacts, and produce systematically lower-quality data in non-European samples.

## Methods

**Data**. iPSYCH2012 is a case-cohort design nested within 1,472,762 individuals born in Denmark between 01-05-1981 and 31-12-2005, with a known mother, alive and residing in Denmark at the end of the first year after birth. Out of 86,189 individuals chosen for genotyping, 57,377 are cases with one or more mental disorders among schizophrenia, autism, attention-deficit/hyperactivity disorder (ADHD), and affective disorder. The cohort is a random sample of 30,000 individuals representative of the national population of Denmark born during the same time period. Genotyping was performed at The Broad Institute, Boston MA, USA with the Infinium Psych Chip v1.0 (Illumina, San Diego CA, USA), using DNA extracted from dried blood spots, obtained from the Danish neonatal screening biobank[37]. Further details on the ascertainment and data generation process of iPSYCH2012 have previously been described[19]. iPSYCH2015i is an extension of iPSYCH2012, nested within 1,717,316 individuals born in Denmark between 01-05-1981 and 31-12-2008, satisfying the same criteria, encompassing 33,345 cases and 15,756 cohort individuals, genotyped on the Illumina Global Screening Array

v2.0 (Illumina, San Diego CA, USA) at Statens Serum Institut, Copenhagen Denmark.

The trio dataset contains 128 parent-offspring trios where the offspring were ascertained for diagnoses of autism or ADHD with both parents born in Denmark, on or after 01-05-1981. Samples were genotyped using both the Infinium Psych Chip v1.0 and the Illumina Global Screening Array v2.0. Information on psychiatric diagnoses was obtained from the Danish national psychiatric central register[38,39], demographic information including age, gender, and parental birthplace was obtained from the Danish civil registration system[40,41].

The Personal Genomes Project—UK (PGP-UK) is an open source initiative aimed at facilitating access to multi-omics datasets for the purpose of gaining insights into biological and medical processes[30] and contains 1100 citizens or permanent residents of the United Kingdom who provided consent after passing a test aimed at educating them on the risks of sharing personal genetic data. DNA was extracted from blood and whole-genome was sequenced using Illumina HiSeq X at an average depth of 15×. The resulting BAM files were deposited to the European Nucleotide Archive (Study identifier: PRJEB17529).

**Genotyping arrays**. The iPSYCH2012 dataset was genotyped using the Infinium Psych Chip v1.0. Per manufacturer details, the array is designed to genotype a total of ~593,260 markers, half of which are haplotype informative tag SNPs as found on the Infinium BeadChip, while the other half are rare markers as found on the Infinium Exome BeadChip. An additional 60,000 markers were added to the manifest based on prior associations with psychiatric disorders. Apart from single nucleotide polymorphisms, short insertions/deletions as well as large copy number variants are ascertained.

The iPSYCH2015i dataset was genotyped using the Illumina global screening array v2.0, which per manufacturer details is designed to genotype ~654,027 markers, mostly single nucleotide polymorphisms, which were especially chosen to deliver high accuracy imputations at common allele frequencies (>0.01) across all 26 sub-populations of the 1000 genomes consortium population dataset. The manifest files for the two iPSYCH arrays overlap at ~179,856 markers.

**Ethical permissions**. Research using iPSYCH and the trio data has been approved by the Danish scientific ethics committee, Danish health authority, and the Danish neonatal screening biobank committee. The consent structure of iPSYCH, as governed by the Act on Research Ethics Review of Health Research Projects in Denmark is exempted from obtaining informed consent from human research participants (https://ipsych.dk/en/data-security/health-research-and-ethical-approval/). PGP-UK has been approved by the University College London scientific ethics committee and informed consent was obtained from all human research participants. All analyses were performed on a secure server within the Danish national life science supercomputing cluster (https://computerome.dtu.dk/) and the Aarhus Genome Data Center (https://genome.au.dk/).

**Genotype QC**. The QC steps prior to phasing are divided into two stages. An initial SNP level QC and a second sample level QC were performed on a subset of individuals of a relatively homogenous genetic origin, as determined through the Danish birth registers and principal components analysis, within the iPSYCH sample.

**Identifying a genetically homogenous sample subset for QC**. Certain steps in the QC process such as tests of Hardy–Weinberg equilibrium, identification of samples with abnormal heterozygosity, etc could be biased by genetic diversity in the dataset. To perform these QC steps in an unbiased manner, we identify a set of samples of a homogeneous genetic origin. To do this, the variant calls from the 1000 genomes phase 3 project[13] were downloaded in VCF format. Within each sub-population of the 1000 genomes dataset, we excluded variants with a minor allele frequency less than 5%, Hardy–Weinberg $p$ values <$10^{-6}$, pairwise correlation ($r^2$) >0.1 within a 1 kb region or were not genotyped on either the Infinium psych chip v1.0 or the Illumina global screening array v2.0. Insertions and deletions and variants in regions with extended linkage disequilibrium were also excluded[42]. The resulting data was merged with iPSYCH2012 and iPSYCH2015i using PLINK[43]. We performed a principal component analysis using the smartpca module of the eigensoft software package[44], the principal components were computed using the 1000 genomes samples, and the iPSYCH2012, iPSYCH2015i were projected into the resulting principal component space. We further utilized the Danish national birth records to identify a set of 47,586 individuals whose parents and both sets of grandparents were born in Denmark. For each sample in our dataset, we calculate the mahalanobis distance of the sample from the multivariate mean of the joint distribution of the first 10 principal components obtained from the 47,586 individuals previously identified. We exclude a sample as an outlier if the distance has a probability less than $5.73 \times 10^{-7}$ under a chi-square distribution with 10 degrees of freedom. This resulted in 120,890 samples classified as inliers being used for QC.

**Aligning to the reference**. All 26 waves of iPSYCH2012, 78 waves of iPSYCH2015i, Trios2012, Trios2015, and the PGP-UK samples were aligned to Haplotype Reference Consortium v1.1 (hereafter, referred to as HRC) using

GenotypeHarmonizer v1.4.20-SNAPSHOT[45]. SNP IDs in the target datasets were harmonized to the SNP IDs in the HRC where a match was found, A/T and G/C SNPs were rescued where possible using linkage disequilibrium information, variants not present in the reference, multi-allelic SNPs and indels were excluded.

**SNP missingness**. Per SNP and sample, missingness was calculated using PLINK 2.0. Genotyping for iPSYCH2012 was performed in 26 waves. We initially excluded variants missing in >5% of samples in each individual wave. The samples were further merged and variants that were either not genotyped in all 26 waves or were found to be missing in ≥ 5% of samples in the merged dataset were further excluded. 344,498 SNPs pass this QC. The genotyping for iPSYCH2015i was performed in 78 waves. We excluded SNPs missing in excess of 5% of the samples in each genotyping wave. Samples were merged across batches and SNPs missing in >5% of samples across the entire cohort were removed. A total of 558,013 SNPs pass missingness filters.

**Differential missingness between cases and controls**. We test for SNPs showing differential missingness between cases and controls of a homogenous genetic origin as described in section 1 using the test-missing option in PLINK. We excluded SNPs that show evidence for differential missingness with an FDR-adjusted *p* value ≤ 0.2. 342,837 SNPs in iPSYCH2012 and 555,131 SNPs in iPSYCH2015i pass this filter.

**Hardy–Weinberg equilibrium**. The individuals of a homogenous genetic origin as derived in section 1 were further subset to include individuals without any disease diagnosis as ascertained from the Danish national patient registers and a test for Hardy-Weinberg equilibrium was performed using the –hardy option in PLINK. We exclude SNPs that fail this test with an FDR-adjusted $p \leq 0.2$. 338,104 SNPs in iPSYCH2012 and 544,308 SNPs in iPSYCH2015i pass this QC.

**Batch artifacts**. Due to the large sample size of iPSYCH, the genotyping for iPSYCH2012 was performed in 26 waves and the genotyping for iPSYCH2015i was performed in 78 waves. To identify markers showing significant batch effects, we performed 26 and 78 logistic regressions in iPSYCH2012 and iPSYCH2015i respectively where samples of a homogenic genetic origin in a particular wave are cases and samples in other waves are controls. For each SNP, we take the minimum of *p* values from all association tests.

The *p* values thus selected do not follow a uniform distribution and the cumulative distribution function of drawing minimums from *n* independent distributions

$$Y = \min(p_1, p_2, \ldots p_m) \tag{1}$$

is given by

$$CDF(y) = p(Y \leq y) = 1 - (1 - y)^m \tag{2}$$

If $p_i$ is the *i*th element in a set of *m* sorted *p* values, the CDF of $p_i$ is given by $\frac{i}{m}$, the *i*th element in a set of m sorted minimum *p* values is given by

$$p_i = 1 - (1 - \frac{i}{m})^{\frac{1}{m}} \tag{3}$$

The qq-plot of observed vs expected *p* values using the above theoretical distribution suggests some inflation. FDR adjustment, defined as the ratio of the expected (under a global null) to observed number of *p* values below a given threshold, can be calculated using the above CDF to estimate the numerator as,

$$p_{fdr} = \frac{m(1 - (1 - p)^m)}{|p < p_i|} \tag{4}$$

We chose an FDR adjusted *p* value cutoff of 0.1 to exclude SNPs, which corresponded to a *p* value of $6.31 \times 10^{-5}$ in iPSYCH2012 and $2.38 \times 10^{-6}$ in iPSYCH 2016. SNPs passing QC filters, iPSYCH2012 = 333,308, iPSYCH2015i = 543,422.

**Minor allele frequency**. A subset of 34,545 individuals in iPSYCH2012 was exome sequenced using the Illumina capture kit on HiSeq machines. QC was performed using HAIL and variant calling was performed in accordance with the GATK best practices. More details on the data processing have previously been described[46]. For these individuals, we calculated genotype concordance between the exome sequencing data and genotypes from the iPSYCH2012 array data using bcftools[47] as shown in Table 1. We observe that the concordance between genotyped and next-generation sequencing datasets drops sharply at minor allele frequencies below 0.001. So, we chose this as a sensible threshold for censoring SNPs. SNPs passing QC filters: iPSYCH2012: 261,551, iPSYCH2015i: 460,445.

**SNP masking**. To evaluate the performance of missing data imputation, we randomly selected 10,000 SNPs that were genotyped on both the Illumina PsychArray v1.0 and the Illumina Global Screening Array v2.0 using the

**Table 1 Concordance between genotypes from Infinium Psych Chip v1.0 and whole-exome sequencing data in a subset of 34,545 individuals in iPSYCH2012.**

| Allele frequency bin | Concordance between genotyping array and exome sequencing data | Number of SNPs |
|---|---|---|
| 0.00001–0.0001 | 0.4085 | 20,701 |
| 0.0001–0.001 | 0.7976 | 30,367 |
| 0.001–0.01 | 0.9676 | 14,145 |
| 0.01–0.1 | 0.9966 | 6795 |
| 0.1–0.5 | 0.999 | 5081 |
| 0.5–1 | 0.9991 | 28 |

sample function in R. These were excluded prior to haplotype estimation. SNPs used for haplotype estimation and imputation, iPSYCH2012: 251,551, iPSYCH2015i: 450,445.

**Abnormal heterozygosity**. Abnormal levels of heterozygosity that cannot adequately be explained by admixture, population structure, or runs of homozygosity could indicate sample contamination. To identify individuals with heterozygosity that cannot be accounted for by population phenomena, we use an approach described by the UK biobank (https://biobank.ctsu.ox.ac.uk/crystal/crystal/docs/genotyping_qc.pdf). Per sample heterozygosity, homozygosity and missingness were calculated using PLINK–het,–homozyg and–missing options respectively. Ancestry-adjusted heterozygosity is computed as the residuals from the model shown below:

$$
\begin{aligned}
H(x) \sim\ & H_0 + PC_1 + PC_2 + PC_3 + PC_4 + PC_1^2 \\
& + PC_2^2 + PC_3^2 + PC_4^2 + PC_1 * PC_2 + PC_2 * PC_3 \\
& + PC_3 * PC_4 + PC_4 * PC_1 + PC_1 * PC_3 + PC_2 * PC_4
\end{aligned} \tag{5}
$$

Where $H(x)$ = observed heterozygosity
$H_0$ = mean heterozygosity/Intercept
$PC_1, PC_2, PC_3, PC_4$ = first four principal components of genetic ancestry
E = residual/ancestry adjusted heterozygosity

We further fit two linear models predicting the observed and ancestry-adjusted heterozygosities from runs of homozygosity calculated using PLINK. Samples are flagged as outliers if the observed and ancestry-adjusted heterozygosity as well as the residuals from the models fit against runs of homozygosity are four standard deviations away from the mean. 166 samples from iPSYCH2012 and 98 samples from iPSYCH2015i failed this quality check and were excluded.

**Sample duplication**. A total of 121 samples were found to be genotyped more than once across the 26 waves in iPSYCH2012. Further, mapping sample identifiers to unique identifiers from the registers yielded 159 sample identifiers in iPSYCH2012 and 25 sample identifiers in iPSYCH2015i mapping to a non-unique identifier in the registry. Two samples from iPSYCH2012 were found to be genotyped again in iPSYCH2015i due to the randomness of ascertainment. In each case, the sample with lower missingness was retained. Six samples in iPSYCH2012 and 1 sample in iPSYCH2015i were genotyped as part of the trios and were excluded. Kinship analysis performed using KING[48] revealed three monozygotic twins in iPSYCH2012 and ten monozygotic twins in iPSYCH2015i. In each case, the case was retained and if both samples were cases, the sample with higher missingness was excluded. For GWAS and PGS analyses, the relatedness cutoff was set to a kinship coefficient <0.0825.

**Sample missingness**. Two samples from the iPSYCH2012 cohort were excluded for excessive missingness (>5%). This left us with 80,876 samples in iPSYCH2012, genotyped at 251,551 loci and 48,974 individuals in iPSYCH2015i, genotyped at 450,445 loci to be used as a backbone for haplotype estimation and missing data imputation.

**Pre-phasing data integration protocols**. We evaluated four different ways of integrating data as shown in Fig. 1.

In the separate protocol (Fig. 1a), samples from iPSYCH2012 and iPSYCH2015i are phased and imputed separately. 124 trio offspring were added to both cohorts. Ten whole genome sequenced samples from the PGP-UK cohort were down-sampled to both the iPSYCH2012 and iPSYCH2015i SNPs that passed QC and merged with both cohorts. This resulted in two cohorts: (1) Cohort2012 (81,022 samples, 251,551 SNPs, 0.1% missingness) which includes iPSYCH2012, trio offspring genotyped on the Infinium Psych Chip v1.0 and ten PGP-UK samples, down-sampled to the Infinium Psych Chip v1.0 variants that pass QC. (2) Cohort2015i (49,120 samples, 450,455 SNPs, 0.31% missingness) which includes the iPSYCH2015i, trio offspring genotyped on the Illumina Global Screening Array

v2.0 and ten PGP-UK samples, down-sampled to the Illumina global screening array v2.0 variants that pass QC.

In the intersection protocol (Fig. 1b), samples from iPSYCH2012 and iPSYCH2015i were merged at the 116,962 QC'ed SNPs present on both iPSYCH arrays. 62 offspring samples were chosen at random from each of the trio datasets genotyped using both iPSYCH arrays and merged to this dataset along with ten PGP-UK samples, down-sampled to the 116,962 common loci. This resulted in the intersection (129,886 samples, 116,962 SNPs, 0.17% missingness) cohort.

In the union protocol (Fig. 1c), samples from iPSYCH2012, iPSYCH2015i were merged with missingness to the 596,028 QC'ed SNP loci, genotyped on either iPSYCH array. To this, 62 samples each from the trio dataset genotyped on both arrays were merged, same as in the intersection. Five PGP-UK samples, each down-sampled to the SNPs present on either genotyping array, were merged resulting in the union cohort (129,886 samples, 596,028 SNPs, 44.54% missingness).

In the two-stage protocol (Fig. 1d), eight sets of phased haplotypes from the Cohort2012 and Cohort2015i obtained in the separate protocol were initially imputed using BEAGLE5.1 in batches of 10,000 samples to the 596,028 QC'ed SNPs genotyped on either iPSYCH array with HRCv1.1 as the reference. Then the two cohorts were merged, retaining the same 62 trio samples from each cohort as chosen in the intersection and union approaches along with five PGP-UK samples from each cohort, forming the two-stage cohort (129,886 samples, 596,028 SNPs, 0% missingness).

All datasets were stored and processed in VCF (http://samtools.github.io/hts-specs/VCFv4.2.pdf) using bcftools[49].

**Phasing**. Cohorts arising from each data integration protocol were phased using three methods and two different parameters, BEAGLE5 (phase-states=280, 560), SHAPEIT4.1.2 (pbwt-depth = 4, 8), EAGLE2.4.1 (Kpbwt = 10000, 20000) with the added aim of benchmarking improvements in accuracy at a higher resolution parameter set at the expense of longer run times and memory requirements. A consensus haplotype set was generated by taking the majority haplotype estimate across the three tools at both the default and higher resolution parameters at each locus within each individual using BEAGLE's consensusvcf (https://faculty.washington.edu/browning/beagle_utilities/utilities.html#consensusvcf) module. The HRCv1.1 dataset, consisting of 64,976 haplotypes[50] was used as the reference panel.

**Imputation**. Due to the cohort sizes, imputations using BEAGLE5.1 at default parameters were carried out in batches of 10,000 samples. Imputations using Impute5 were carried out in multiple steps, an initial step to define chunks of the region to impute, followed by imputation, and finally, concatenation of chunks using bcftools. For imputations using Minimac4, the reference HRC haplotypes were converted to m3vcf format using Minimac3, followed by imputation using Mimiac4. All imputations used default parameter settings for the tools. Imputed dosage (DS) for an individual at a bi-allelic locus is calculated as $DS = p(RA) + 2*p(AA)$ where $p(RA)$ is the genotype probability corresponding to the presence of one alternate allele (A) and one reference allele (R) as per the reference panel and $p(AA)$ corresponding to the genotype probability of the presence of two copies of the alternate allele.

**Phasing accuracy**. Phasing accuracy was evaluated by calculating SER in the trio offspring at the QC'ed heterozygous SNPs common to both iPSYCH arrays. A switch error arises from an inconsistency between the computationally assigned phase and the phase observed by mendelian transmission with knowledge of parental haplotypes. SER is the number of such switches divided by the total possible switches[51]. The code for SER calculation has previously been used[9] and available on Github (https://github.com/odelaneau/switchError).

**Imputation accuracy**. Imputation accuracy within iPSYCH was calculated as the squared Pearson correlation coefficient ($r^2$) between true genotypes and imputed dosages within different minor allele frequency (MAF) bins (MAF as measured in HRCv1.1) at each of the 10,000 SNPs masked prior to phasing. Imputation accuracy within PGP-UK was calculated as the $r^2$ between true genotypes obtained from multi-sample variant calling using samtools[49] and imputed dosages in eight MAF bins at 6,517,513 loci that were genotyped on neither iPSYCH array. The code is available on GITHUB (https://github.com/vaqm2/impute_paper/blob/main/truth_vs_impute_2021_02_24.pl). To evaluate variations in imputation accuracy by ancestral origin, $r^2$ was calculated within iPSYCH samples, grouped according to the country of birth of both parents according to the Danish civil register[40,41].

**Phenotype simulations**. To evaluate the impact of whole genome imputation on PGS, a quantitative trait for 129,850 iPSYCH individuals was simulated using GCTA[52] version 1.92.1beta6, with a heritability of 0.5 and the 10,000 masked SNPs as causal loci with effect sizes drawn from a standard normal distribution.

**Association tests**. To evaluate the presence of batch artifacts in each protocol we conducted multiple GWAS with iPSYCH cohort membership (iPSYCH2012 vs iPSYCH2015i) as the outcome using the glm module of PLINKv2.00a2LM 64-bit Intel (10 Nov. 2019)[53]. As a baseline, we performed the GWAS using true values of 10,000 masked genotypes as explanatory variables. Subsequently, GWAS were performed comparing allele frequencies from true genotypes in one cohort to imputed dosages in the other and imputed dosages in both, across all four data integration protocols (separate, union, intersection, two-stage). Tests were restricted to iPSYCH individuals without mental disorders (i.e., a random sample of psychiatric controls), of a homogenous genetic origin based on principal component analysis using Eigenstrat[44], and pruned for relatedness beyond the third degree using kinship coefficients estimated by KING[48]. The overall inflation of test statistics above the null was evaluated using the genomic inflation factor which compares the median of the chi-square test statistic obtained from each GWAS to the expected median of a chi-square distribution with 1 degree of freedom.

**Polygenic scores**. To calculate the PGS for each individual, $j$, we initially performed a GWAS within unrelated individuals of a Danish genetic background as ascertained by principal component analysis (Supplementary S1.1, S1.3.2) from the iPSYCH2012 and iPSYCH2015i cohorts using PLINK2 –glm option with the simulated continuous phenotype as outcome. The per SNP effect sizes from each iPSYCH cohort was in turn used as the discovery set ($\beta_i$) to calculate PGS for individuals in the other cohort as follows:

$$PGS_j = \sum_{i=1}^{m} \beta_i X_{ij} \qquad (6)$$

where $m$ is the total number of SNPs (10,000 masked SNPs), $\beta_i$ is effect for SNP $i$ in the discovery GWAS, $X_{ij}$ is the imputed dosage or best guess genotype count of effect alleles for individual j at SNP i. Variance explained by PGS was calculated by fitting two linear models using the function, lm in R. The simulated trait value is the outcome, individual PGS is the sole explanatory variable in one model, while individual PGS, age, gender, and first 10 principal components of genetic ancestry are explanatory variables in the second model. Variance in the simulated trait value explained by PGS is the difference between the correlation coefficients ($R^2$) between the two models. We restrict the analysis to 67,587 individuals from iPSYCH2012 and 41,069 individuals from iPSYCH2015i with parents and both sets of grandparents born in Denmark and clustering with the CEU (Utah residents with Northern and Western European ancestry) and GBR (British in England and Scotland) populations of the 1000 genomes phase 3 dataset in principal component analysis.

**Reporting summary**. Further information on research design is available in the Nature Research Reporting Summary linked to this article.

## Data availability

Source data underlying figures are provided in Supplementary Data 1–9. The haplotype reference consortium can be downloaded from the European Genome-Phenome archive with the study identifier: EGAD00001002729. Personal genomes project UK datasets are publicly available for download at https://www.personalgenomes.org.uk/data/ and through the European nucleotide archive with the accession number: PRJEB17529. In accordance with the Danish law and consent permissions governing iPSYCH, the individual-level genotype data from the iPSYCH and trios datasets cannot be shared publicly. The data, hosted on Genome DK is available for researchers involved with the consortium. Further information on iPSYCH data security can be found here: https://ipsych.dk/en/data-security, the ethical approval statement is available here: https://ipsych.dk/en/data-security/health-research-and-ethical-approval. Further queries regarding the consortium can be directed to the concerned research committee here: https://ipsych.dk/en/contact.

## Code availability

The phasing tools are available from the project webpages. Beagle5 and its associated utilities, ConformGT and ConsensusVCF can be downloaded from https://faculty.washington.edu/browning/beagle/b5_1.html, http://faculty.washington.edu/browning/conform-gt.html, and https://faculty.washington.edu/browning/beagle_utilities/utilities.html#consensusvcf. Eagle2.4.1 can be downloaded from http://data.broadinstitute.org/alkesgroup/Eagle/downloads/. Shapeit4.1.2 can be downloaded from https://odelaneau.github.io/shapeit4/. Switch errors in haplotype estimation were calculated using code available on GitHub at https://github.com/SPG-group/switchError. Imputation accuracies were estimated using code available on GitHub at https://github.com/vaqm2/impute_paper. Bcftools (version 1.9) can be obtained from https://github.com/samtools/bcftools and PLINK2 can be downloaded from https://github.com/chrchang/plink-ng. Genotype harmonizer v1.4 is available on GitHub at https://github.com/molgenis/systemsgenetics/tree/master/Genotype-Harmonizer.

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

## Acknowledgements

Vivek Appadurai is supported by the Lundbeck Foundation postdoctoral grant: R380-2021-1465. Andrew Schork is supported by the Lundbeck Foundation fellowship: R335-2019-2318. The iPSYCH team was supported by grants from the Lundbeck Foundation (R102-A9118, R155-2014-1724, and R248-2017-2003), NIMH (1R01MH124851-01 to A.D.B.) and the Universities and University Hospitals of Aarhus and Copenhagen. The Danish National Biobank resource was supported by the Novo Nordisk Foundation. High-performance computer capacity for handling and statistical analysis of iPSYCH data on the GenomeDK HPC facility was provided by the Center for Genomics and Personalized Medicine and the Centre for Integrative Sequencing, iSEQ, Aarhus University, Denmark (grant to A.D.B.).

## Author contributions

V.A. contributed to study conception, experimental design, coding, data cleaning, data analysis, and manuscript writing. J.B.G. is responsible for the genotyping and sequencing of iPSYCH and initial quality control. M.D.K. offered vital feedback on experimental design and manuscript writing. A.R. contributed to data analysis. A.B. and A.I. contributed to the study conception and experimental design. P.B.M., A.D.B., T.W., O.M., M.N., D.M.H. conceived the concept of iPSYCH conception, directed data collection, and

generation, and obtained funding. O.D. is the author of Shapeit4 and contributed to the study conception, experimental design, and code for computing switch error rates. A.J.S supervised the project, and contributed to the conception of the study, experimental design, data cleaning, and manuscript writing. All authors read and acknowledged the manuscript.

## Competing interests

The authors declare no competing interests.
