## [Peer Review File · Communications Biology]

Reviewers' comments:

Reviewer #1 (Remarks to the Author):

In this manuscript, the authors assessed how different data integration strategies would affect phasing and imputation errors, and further affect the polygenic scores (PGS). Leveraging biobank scale data, the authors designed four data integration protocols and performed thorough experiments to achieve this goal. The manuscript is well written, with clearly conveyed messages, but there are still some questions need to be addressed to make it in a better shape.

1. (Major) As the authors mentioned on page XVI lines 360-361, I believe the goal that most people perform genotype imputation is to attain the most accurate missing data imputation rather than phasing alone, which makes me think that imputation method/software may be more important than phasing method/software. Did the authors try comparing the imputation errors across different pipelines (e.g., BEAGLE5, minimac4, IMPUTE5)?
2. (Major) The authors constructed PGS by taking the simulated true effects of each true causal SNP, which is interesting to see the attenuated variance explained using imputed data, but it's not realistic in real-life applications. We can never know what the true causal SNPs and what the true effects are. I suggest the authors perform GWAS first and use the GWAS results only to construct PGS, to mimic a more realistic situation.
3. (Major) Related to the comments above, it would also be interesting to look at the genomic control coefficient in the simulated trait GWAS, as investigations of how phasing and imputation errors would affect GWAS.
4. (minor) Figure 2 (b)(c) are not informative to me. The letter notations on plots overlap with each other and are hard to distinguish. The authors should think of a better visualization.

Reviewer #2 (Remarks to the Author):

This is a substantial and comprehensive study that examines the impact of data integration strategies on the accuracy of genotype phasing and imputation. The data and conclusions will be useful to all researchers who are analyzing multiple cohorts that have been genotyped on different arrays. The study is well-designed and the paper is well written. I have only minor comments.

Minor Comments.

P. 16, The impact of marker density on imputation accuracy is particularly strong because the 116,962 SNPs in the intersection of the arrays is much too low for LD-based genome-wide analysis in outbred human populations. Even the 251,551 SNPs on the smaller array is rather low. Do you think the effect of marker density on imputation accuracy would be smaller if the two arrays had 500k and 1M markers with 300k markers in common?

Introduction: "genome wide association studies (GWAS)^{1,2} require hundreds of thousands of participants to identify trait-associated loci." This needs qualification. 100,000s of participants are needed to identify *some* trait-associated loci. Some associated loci can be identified with much smaller sample sizes.

P. 6, Can you provide some additional information about the two SNP arrays. For example, the number of markers on each array, and how the markers were selected.

Supplemental Data, P. 34, "Pairwise $r^2 > 0.1$ in a 1kB region". Does this mean $r^2 > 0.1$ with all markers in a 1 kb radius of a focal marker?

P. 37, " $CDF(Y) = p(Y \leq y) = 1 - 1(1 - y)^n$ ". As written, the second "1" is redundant.

P. 38, Just checking to confirm the superscript is supposed to be "1/n", and not "n".

P. 38, Can you include a reference for the " $p_{\text{fdr}} = m - (1 - \pi)n / \sum(p < \pi)$ " formula. Is " p_{fdr} " the FDR-adjusted p-value? Where does the chosen FDR come into this formula?

Math equations. If you are using MS Word, you could consider using MS Word's built-in equation editor to improve the typesetting of some or all of the math formulae in the Supplemental Information.

Reviewer 1 asks:

1. (Major) As the authors mentioned on page XVI lines 360-361, I believe the goal that most people perform genotype imputation is to attain the most accurate missing data imputation rather than phasing alone, which makes me think that imputation method/software may be more important than phasing method/software. Did the authors try comparing the imputation errors across different pipelines (e.g., BEAGLE5, minimac4, IMPUTE5)?

Our response:

As the primary aim of our study is to investigate the optimal choice of data integration scenarios, we chose to focus on the impact of phasing tools for the following reasons:

1. If imputation is thought of as a two-step process, involving an initial pre-phasing step, followed by an imputation step, in most imputation pipelines, its the phasing tools that encounter the peculiarities arising from the various data integration scenarios, such as missingness in the data and the sparsity of study markers. In case of missingness, the phasing tool performs an initial imputation step, prior to estimating haplotypes using the haplotype information present in the reference and study datasets. Therefore, we felt the biggest impact of imperfections in data integration will be encountered and solved by the phasing tools and there has been no research on how different tools handle them.
2. The prevailing literature in method development for imputations (Rubinacci et al., 2020), suggests that different imputation tools have reached a saturation in terms of accuracy and therefore compete in terms of computational efficiency.
3. Lastly, testing four data integration scenarios, across four different phasing methods at two different parameter settings each has given us a total of 32 different datasets and adding an additional permutation of imputation tools would have expanded the scope of our study beyond reasonable time and computational possibilities.

With these concerns in mind, we proceeded to test the performance of Minimac4, Impute5 and Beagle5.1 over the masked loci on chromosome 22 across all four data integration protocols with haplotypes pre-phased using Beagle5.1, the most consistent phasing tool. The results presented in Supplementary Figure 5 and Supplementary Table 11 show that Beagle5.1 was better across three of the four data integration scenarios. We note that relative to phasing methods we do see

less variability in imputation quality in relation to the choice of imputation methods across the various data integration scenarios.

We now refer to these analyses in the methods section, lines 514-517:

Imputations using Impute5 were carried out in multiple steps, an initial step to define chunks of the region to impute, followed by imputation and finally, concatenation of chunks using bcftools. For imputations using Minimac4, the reference HRC haplotypes were converted to m3vcf format using Minimac3, followed by imputation using Mimiac4.

and results sections, lines 219-222:

Previous studies have suggested lower variability in imputation accuracy as a function of imputation (versus phasing) tools³¹, a result we confirmed in limited a comparison of BEAGLE5.1⁸, Minimac4³² and Impute5³¹ is presented in Supplementary section S9, Supplementary Figure 5 and Supplementary Table 11.

of the main paper and Supplementary section S9., lines 278-303

Supplementary Figure 5. Comparison of imputation accuracies from three different imputation tools across the four different data integration protocols. Pre-phasing was done using BEAGLE5.

Reviewer 1 asks:

2. (Major) The authors constructed PGS by taking the simulated true effects of each true causal SNP, which is interesting to see the attenuated variance explained using imputed data, but it's not realistic in real-life applications. We can never know what the true causal SNPs and what the

true effects are. I suggest the authors perform GWAS first and use the GWAS results only to construct PGS, to mimic a more realistic situation.

Our response:

This is true, that we never know the true causal SNP, nor the true causal effects. Our intention was to limit one source of variability (estimation error in betas) to make a point that we feel has not received enough attention in the published literature (estimation error in genotypes) with respect to PGS. Given the variability we observe in imputation quality by genetic ancestry, this could be an important, and less appreciated, contributor to the failure of PGS to predict across populations. We agree with the reviewer that this was an artificial demonstration and a valid criticism. We have therefore extended our PGS analysis module to compare 3 sources of error: error in imputed genotypes in the test sample, error in imputed genotypes in the training sample (which add error to estimated betas), and error in estimated betas in the training sample due to sampling variance. We utilized the independent sample ascertainment design of iPSYCH, performed a GWAS in iPSYCH2012 with the simulated continuous trait as the outcome and used SNP effects from this GWAS to calculate polygenic scores in iPSYCH2015i and vice-versa. We believe this mimics the more realistic situation suggested by the reviewer and these changes are reflected in the methods section, lines 574-578:

To calculate the polygenic scores (PGS) for each individual, j , we initially performed a GWAS within unrelated individuals of a Danish genetic background as ascertained by principal component analysis (Supplementary S1.1, S1.3.2) from the iPSYCH2012 and iPSYCH2015i cohorts using PLINK2 `-glm` option with the simulated continuous phenotype as outcome. The per SNP effect sizes from each iPSYCH cohort was in turn used as the discovery set (β_i) to calculate PGS for individuals in the other cohort as follows:

Results section, lines 244-285:

Imputations can contribute to sub-optimal predictive performance of PGS by introducing measurement error in the genotypes used in the discovery GWAS as well as in the target genotypes used for scoring. To empirically compare attenuation in predictive performance due to imputation errors at each stage, we utilized the independent sample ascertainment of the two iPSYCH cohorts, employing

iPSYCH2012 as discovery cohort while predicting in iPSYCH2015i and vice-versa. To estimate the impact of imputation errors in discovery GWAS, we performed a GWAS using PLINK with a simulated continuous phenotype as the outcome, regressed against the imputed dosages at the 10,000 masked SNPs in one iPSYCH cohort and calculated PGS by summing the product of the obtained SNP effect sizes with true genotypes at the 10,000 masked loci in the other iPSYCH cohort. To estimate the impact of imputation errors in target genotypes, we performed GWAS using true genotypes in one iPSYCH cohort, while calculating PGS in the other cohort as the running sum of the product of per SNP effect sizes with imputed dosages at the 10,000 masked SNPs (See methods for more details).

The results presented in Figure 4 suggest that the impact of imputation errors is more severe when present in target genotypes used for scoring than when they are present in the genotypes used for discovery GWAS. When using imputed dosages from iPSYCH2012 in a discovery GWAS and scoring using true genotypes in iPSYCH2015i, the only noticeable attenuation in the variance explained by PGS occurs in the *intersection* protocol, diminishing to 0.26 as compared to 0.27 in the ideal scenario, where both GWAS and scoring use true genotypes. On the other hand, when the discovery GWAS used true genotypes and scoring was done using imputed dosages, there is an attenuation in variance explained by PGS across all four imputation protocols, increasing in severity from the separate protocol ($r^2 = 0.26$) to the intersection protocol ($r^2 = 0.23$). This demonstration of reduced predictive capability of PGS in presence of imputed genotypes is in line with animal breeding studies²⁴ and highlights an additional challenge when projecting PGS across populations.

Another application of PGS is to prioritize individuals in top quantiles of a PGS distribution for monitoring and intervention. To investigate the effect of imputation accuracy on such an application, individuals in iPSYCH2015i were ranked into percentiles for the simulated continuous trait based on a PGS calculated using SNP effects with iPSYCH2012 as reference GWAS, true genotypes and imputed dosages across all four data integration protocols, in turn accounting for the genetic dose. The results (Figure 4b) are consistent with prior work²⁵ showing a discrepancy in individual rank that is higher in the middle percentiles and lower in the more actionable top percentiles of the PGS distribution. The discordance in individuals in the top percentiles between PGS constructed by true genotypes and imputed dosages is, however, much higher than the 5% previously reported. The overlap in the proportion of individuals ranked in the top 5 percentiles of PGS using true genotypes and imputed dosages is highest in both cohorts

when employing either the *separate* (0.85%) or *twostage* (0.86%) protocol and lowest when using the *intersection* (0.75%) protocol. Our results suggest that using imputed genotypes for the discovery GWAS stage of constructing a PGS might be more robust than when incorporating imputed genotypes into the scoring stage. The attenuation in predictive performance, discordance of individuals in the actionable percentiles of PGS follow the quality of imputed dosages obtained from each data integration protocol, showcasing the importance of optimal data integration for genetic prediction and may have special relevance for cross-ancestry PGS applications, where errors due to imputation are systematically larger.

figure 4, figure description (lines 632-642)

Figure 4. The performance of polygenic scores is attenuated when using imputed data, which affects both population level measures such as variance explained in a simulated phenotype and in individual level metrics, such as rank concordance when compared to true genotypes.

[a] Shows the attenuation in variance explained by a PGS when error introduced by using imputed dosages resides in either the discovery stage or in scoring. The green bars are the reference, where true genotypes were used for both discovery and scoring. Red bars indicate that imputed dosages were used in discovery, while the blue bars indicate that imputed dosages were used for scoring. [b] Shows the rank concordance between individuals in iPSYCH2015i ranked according to their PGS calculated using true genotypes on X-axis and imputed dosages from each data integration protocol, across the four figure panels on the Y-axis.

and supplementary table 4 concerning PGS. Briefly, we observed the impact of uncertainty in imputations affects the predictive accuracy a lot more when it is used in the scoring stage, leading to an attenuation in variance explained across all four data integration protocols, as compared to when used in the discovery GWAS. We have now highlighted the importance of further work on all three sources of variability in a PGS to better explore cross-ancestry prediction.

Reviewer 1 asks:

3. (Major) Related to the comments above, it would also be interesting to look at the genomic control coefficient in the simulated trait GWAS, as investigations of how phasing and imputation errors would affect GWAS.

Our response:

We agree with the reviewer that this is an interesting analysis. We calculated the genomic control coefficients from the GWAS performed on the simulated phenotype using iPSYCH2012, 2015i and the combined cohort, iPSYCH2015 across all four data integration protocols. These results are presented in Supplementary section S10, Supplementary figure S6 and Supplementary Table 6.

Supplementary Figure 6. The Genomic control coefficient, as a measure of the loss of power and capability to recover the polygenic signal, calculated from the association statistics obtained by regressing the simulated continuous phenotype against the true genotypes and imputed dosages across all four data integration protocols in iPSYCH2012, 2015i and the combined cohort at the 10,000 masked causal markers used in the simulations.

Reviewer 1 asks:

4. (minor) Figure 2 (b)(c) are not informative to me. The letter notations on plots overlap with each other and are hard to distinguish. The authors should think of a better visualization.

Our response:

We agree that the visualization is a little hard to read, primarily because the performance of different phasing tools in certain data integration scenarios is very much alike, leading to overlapping observations. We have remade the plot with more conventional shapes and use different colors, line types to distinguish data integration protocols and phasing tools. These updated plots are presented in Figure 2, figure description (lines 616-618).

b] Shows the imputation accuracy (r^2) within each data integration protocol (shape), and choice of phasing tool (color) at different minor allele frequency bins at the 10,000 SNPs common to both genotyping arrays that were masked prior to phasing.

Reviewer 2 asks:

1. P. 16, The impact of marker density on imputation accuracy is particularly strong because the 116,962 SNPs in the intersection of the arrays is much too low for LD-based genome-wide analysis in outbred human populations. Even the 251,551 SNPs on the smaller array is rather low. Do you think the effect of marker density on imputation accuracy would be smaller if the two arrays had 500k and 1M markers with 300k markers in common?

Our response:

We agree with the reviewer that the choice of arrays used for iPSYCH makes this a peculiar scenario where the overlap is extremely low and hence, the *intersection* performs poorest of the data integration protocols we have tested. As we refer to in the introduction, the UK Biobank study, which uses the Axiom and BiLEVE arrays with a very high marker overlap employs an intersection strategy. A similar issue might also arise if, unlike the two iPSYCH cohorts, one cohort is a magnitude smaller than the other. In such a scenario, a *twostage* protocol and the associated bioinformatics complexity might become essential to leverage the full study sample size for more accurate phasing in the smaller of the cohorts. Another scenario, where one of the two arrays is a full subset of the other might also favor a *twostage* protocol with one cohort acting as a reference for the smaller array for initial imputation to a union marker set. However, there is no empirical way to answer at what threshold one protocol might fare better than the other without doing the necessary bioinformatics comparisons, such as experimentally iterating the marker overlap and sample sizes of two or more cohorts. We believe this is an even more comprehensive and generalizable study design and a possible extension of our work. We have amended the discussion section, lines 373-380 to include an expanded discussion of this issue and a caveat to our recommendations.

It is important to acknowledge caveats associated with this study, that it studies the outcomes when the choice of genotyping arrays are *very dissimilar* and the two cohorts being studied *do not* differ by an order of magnitude. Therefore, there can exist scenarios, where a data integration protocol such as *union* or

twostage might prove necessary to enhance the number of informative study haplotypes if one of the two cohorts is much smaller than the other. Similarly, if the overlap of SNPs is a magnitude higher, it might prove beneficial to prioritize the dramatic reduction in batch effects and simplicity of an *intersection* protocol. Such empirical comparisons can only be made by iterating across sample sizes and marker densities, across cohorts, in a more broadly orchestrated study.

Reviewer 2 asks:

2. *Introduction: “genome wide association studies (GWAS)^{1,2} often require hundreds of thousands of participants to deliver power to detect trait-associated loci in complex, highly polygenic traits.” This needs qualification. 100,000s of participants are needed to identify *some* trait-associated loci. Some associated loci can be identified with much smaller sample sizes.*

Our response:

We note the concerns of the reviewer and amended the introduction section in the main text to say, “often employ” instead of a definite verb, “require” in lines 52-53:

A recent appreciation for the polygenic nature of complex traits, with several small-effect risk loci scattered throughout the genome has revealed that genome wide association studies (GWAS)^{1,2} **often employ** hundreds of thousands of participants to identify trait-associated loci.

Reviewer 2 asks:

3. *P. 6, Can you provide some additional information about the two SNP arrays. For example, the number of markers on each array, and how the markers were selected.*

Our response:

We have added a description of the two iPSYCH arrays with manufacturer’s descriptions and marker overlap from the manifest files in the subsection “Genotyping Arrays” under Materials and Methods section, lines 419-436:

Genotyping Arrays

Infinium Psych Chip v1.0:

The iPSYCH2012 dataset was genotyped using the Infinium Psych Chip v1.0. Per manufacturer details, the array is designed to genotype a total of approximately 593,260 markers, half of which are haplotype informative tag SNPs as found on the Infinium BeadChip, while the other half are rare markers as found on the Infinium Exome BeadChip. An additional 60,000 markers were added to the manifest based on prior associations to psychiatric disorders. Apart from single nucleotide polymorphisms, short insertions/deletions as well as large copy number variants are ascertained.

Illumina Global Screening Array v2.0:

The iPSYCH2015i dataset was genotyped using the Illumina global screening array v2.0, which per manufacturer details is designed to genotype ~ 654,027 markers, mostly single nucleotide polymorphisms, which were especially chosen to deliver high accuracy imputations at common allele frequencies (> 0.01) across all 26 sub-populations of the 1000 genomes consortium population dataset. The manifest files for the two iPSYCH arrays overlap at ~ 179,856 markers.

Reviewer 2 asks:

- 4. Supplemental Data, P. 34, "Pairwise $r^2 > 0.1$ in a 1kB region". Does this mean $r^2 > 0.1$ with all markers in a 1 kb radius of a focal marker?*

Our response:

Yes, the markers were pruned using PLINK1.9 –indep-pairwise module with a window of 1000 base pairs and r^2 of 0.1.

Reviewer 2 asks:

- 5. P. 37, "CDF(Y) = $p(Y \leq y) = 1 - 1(1 - y)^n$ ". As written, the second "1" is redundant.*

6. P. 38, Just checking to confirm the superscript is supposed to be “1/n”, and not “n”.
7. P. 38, Can you include a reference for the “ $p_{\text{fdr}} = m - (1 - p_i)n/\sum(p < p_i)$ ” formula. Is “ p_{fdr} ” the FDR-adjusted p-value? Where does the chosen FDR come into this formula?
8. Math equations. If you are using MS Word, you could consider using MS Word’s built-in equation editor to improve the typesetting of some or all of the math formulae in the Supplemental Information.

Our response:

We have provided the correct equations with a better equation editor in Supplementary section S1.2.5.

The p-values thus selected do not follow a uniform distribution and the cumulative distribution function

of drawing minimums from n independent distributions, $Y = \min(p_1, p_2, \dots, p_m)$ is given by

$$CDF(y) = p(Y \leq y) = 1 - (1 - y)^m$$

If p_i is the i^{th} element in a set of m sorted p-values, the CDF of p_i is given by $\frac{i}{m}$, the i^{th} element in a set of m sorted minimum p-values is given by

$$p_i = 1 - \left(1 - \frac{i}{m}\right)^{\frac{1}{m}}$$

The qq-plot of observed vs expected p-values using the above theoretical distribution suggests some inflation.

FDR adjustment, defined as the ratio of the expected (under a global null) to observed number of p-values below a given threshold, can be calculated using the above CDF to estimate the numerator as,

$$P_{fdr} = \frac{m (1 - (1 - p)^m)}{|p < p_i|}$$

This equation for the false discovery rate we use is taken as a generalization of the Benjamini and Hochberg FDR procedure. The B-H FDR is estimated as the expected number of false positive tests divided by the total number of positive tests (tests with p less than some threshold). Given a p-value threshold, the first (expected number) can be taken as the number of p-values expected to be below the threshold under the global null hypothesis (here, that all GWAS are null) and is defined generally as the total number of tests, m, times the CDF of the global null at the chosen p-value threshold. We define the CDF of global null as the CDF of the minimum of multiple uniform distributions (the marginal uniform, being the null of a single test) as in the supplementary document. For the observed, we count the number below the threshold.

We do not know of a citation for this as we defined it ourselves as an *ad hoc* solution for selecting “plausibly good quality SNPs” on account of their p-value association with multiple batches or waves. Here, we face a trade-off between being too liberal (including bad data) and too conservative (throwing out signal). This procedure has had good face validity for us when looking at distributions of SNP quality metrics in our previous work (Schork et al 2019, Supplementary Figure 20) and the CDF proposed fits simulated data for minimum p-values from multiple uniform distributions.

References

- Rubinacci, S., Delaneau, O., & Marchini, J. (2020). *Genotype imputation using the Positional Burrows Wheeler Transform*. In *Cold Spring Harbor Laboratory* (p. 797944). <https://doi.org/10.1101/797944>
- Schork, A.J., Won, H., Appadurai, V. et al. *A genome-wide association study of shared risk across psychiatric disorders implicates gene regulation during fetal neurodevelopment*. *Nat Neurosci* **22**, 353–361 (2019). <https://doi.org/10.1038/s41593-018-0320-0>

REVIEWERS' COMMENTS:

Reviewer #1 (Remarks to the Author):

I really appreciate the efforts that the authors made to respond to my comments. All my concerns have been resolved and I have no further comments.

Reviewer #2 (Remarks to the Author):

The revision addresses my previous comments. I have no further comments.